# Binge Eating and Obesity Differentially Alter the Mesolimbic Endocannabinoid System in Rats

**DOI:** 10.3390/ijms26031240

**Published:** 2025-01-31

**Authors:** Florian Schoukroun, Karin Herbeaux, Virginie Andry, Yannick Goumon, Romain Bourdy, Katia Befort

**Affiliations:** 1Laboratoire de Neurosciences Cognitives et Adaptatives (LNCA), Université de Strasbourg, UMR 7364, CNRS, 12 Rue Goethe, 67000 Strasbourg, France; florians@wustl.edu (F.S.); herbeaux@unistra.fr (K.H.); 2Institut des Neurosciences Cellulaires et Intégratives (INCI), UPR 3212, CNRS, 8 Allée du Général Rouvillois, 67000 Strasbourg, France; virginie.andry@unistra.fr (V.A.); ygoumon@unistra.fr (Y.G.)

**Keywords:** obesity, binge eating disorder, sucrose, fat, endocannabinoid, RMTg, tVTA

## Abstract

Binge eating disorder (BED) is characterized by the rapid overconsumption of palatable food in a short amount of time, often leading to obesity. The endocannabinoid system (ECS), a system involved in palatable food intake, is highly expressed in reward-related brain regions and is involved in both obesity and BED. This study investigated differences in ECS expression between these conditions using male Wistar rats exposed to specific regimen over six weeks: a non-access group (NA) with a standard diet, a continuous access group (CA) with free-choice high-fat high-sugar (fcHFHS) diet modeling obesity, and an intermittent access group (IA) with intermittent fcHFHS access modeling BED. Food intake was measured, and brain tissues from the nucleus accumbens (NAc), dorsal striatum (DS), ventral tegmental area (VTA), and rostromedial tegmental nucleus (RMTg) were analyzed for ECS expression using qPCR and mass spectrometry. We identified differential ECS expression across palatable food access groups, with variations depending on the brain region (striatal or mesencephalic). Correlation analyses revealed ECS dysregulations dependent on the type (fat or sucrose) and quantity of palatable food consumed. Comparative network analysis revealed co-regulation patterns of ECS-related genes with specific signatures associated with each eating pattern, highlighting RMTg as a key region for future research in eating behavior.

## 1. Introduction

### 1.1. Pathologies and Animal Models

The overconsumption of palatable food rich in fat and sugar is increasing in modern Western societies [1,2] and can lead to maladaptive, addictive-like feeding behaviors. Binge eating disorder (BED) is commonly associated with obesity, along with significant somatic and mental health comorbidities, representing substantial health risks [3,4]. Treatment options remain limited, and relapse rates in patients suffering from BED or obesity are high [5]. The causes and consequences of recurrent food overconsumption remain unclear, and animal models are essential tools for investigating the underlying cerebral mechanisms. The most commonly consumed foods during binge episodes are high-fat and high-sugar items, and several diet paradigms have been developed in rodent to study the two pathologies [3,6]. High-fat diets (HFD) with enriched pellets are widely used to develop obesity in rodents and can be compared to standard food or high sucrose diets (HSD). Interestingly, the free-choice high-fat high-sucrose paradigm (fcHFHS), initially developed by La Fleur et al. [7], combines access to fat and sucrose in distinct forms (liquid sucrose and solid fat), reflecting the diversity of modern human diets and serving as a valuable preclinical model for obesogenic studies [8]. For BED models, factors such as restriction–refeeding cycles and stress are manipulated to induce binge-like eating behaviors. Notably, preclinical studies demonstrate that intermittent access to palatable food, as opposed to continuous access, reliably triggers binge-like intake [9], providing a robust framework to compare obesity and BED under distinct feeding patterns in rodents.

### 1.2. Mesolimbic Brain Regions Involved in Feeding Disorders

Alterations in reward-related brain regions, including the nucleus accumbens (NAc), dorsal striatum (DS), and ventral tegmental area (VTA) have been reported in both obesity and BED [10]. These mesolimbic structures play a central role in regulating reward processing and palatable food consumption [11,12,13,14]. Additionally, the rostromedial tegmental nucleus (RMTg), a GABAergic structure, exerts inhibitory control over VTA dopamine neurons, modulating reward signaling and addictive behaviors associated with substances, such as cocaine, opioids, alcohol, and cannabinoids [15,16,17,18,19,20]. Notably, we recently showed that this inhibitory control influences palatable food overconsumption and relapse [21]. These findings highlight the involvement of mesostriatal brain regions in BED and obesity, with the RMTg potentially playing a key modulatory role through its projections to the VTA. However, further research is required to determine whether the same neurocircuits underlie both conditions.

### 1.3. The Endocannabinoid System ECS

BED and obesity broadly impact neurotransmission systems. In addition to the classic contribution of DA, the endocannabinoid system (ECS) plays a major role in food intake through both central and peripheral mechanisms [22,23]. The ECS is involved in BED [24], obesity [25], and relapse phenomenon [26]. This neuromodulator system comprises lipid-based endocannabinoids, such as anandamide (AEA) and 2-arachidonoylglycerol (2-AG), their synthesis (NAPE-PLD/DAGLα) and degradation (FAAH/MAGL) enzymes, and two well-characterized receptors, the cannabinoid receptors CB1 and CB2. Another receptor, the orphan GPR55, is also a target of cannabinoid compounds [22]. The ECS is largely expressed in reward brain regions [27] as well as in the RMTg [28], where it influences DA signaling, affecting both the processing of reward-related signals and addictive-like behavioral outcomes [28,29].

Recent evidence from our laboratory and others suggests that ECS plasticity may contribute to obesity and BED [24,30]. However, ECS regulation remains challenging to disentangle, as studies report conflicting results, particularly concerning CB1 receptor levels. For instance, fat bingeing is associated with decreased CB1 expression [31,32,33]; whereas, sucrose bingeing increases CB1 levels [34,35,36,37]. These findings underscore the need to further investigation into ECS modulations in both BED and obesity, especially in the RMTg, to determine whether similar or distinct mechanisms contribute to the development or maintenance of these pathologies. It is important to note that the modulation of the ECS may also be impacted by peripheral signals, as this system is altered in various peripheral pathologies, including metabolic diseases, such as obesity, insulin resistance, type 2 diabetes, and cardiovascular diseases [38].

To clarify these mechanisms, we used the fcHFHS paradigm to compare the effects of a continuous (CA, obesity) and an intermittent (IA, BED) access to palatable food over 6 weeks on ECS expression in male Wistar rats. We analyzed striatal and mesencephalic brain regions using qPCR and mass spectrometry, followed by network analysis to identify the co-regulation patterns of ECS-related gene expression.

## 2. Results

### 2.1. The fcHFHS Diet Induced Binge-like and Obesogenic-like Phenotypes

Rats were exposed either to standard diet (NA), continuous (CA) access, or intermittent (IA) access to a fcHFHS diet for six weeks (cohort A, qPCR analysis) or five weeks (cohort B, mass spectrometry analysis) (Figure 1A). During the 2 h access period, IA rats consumed significantly more sucrose and fat compared to CA rats (Figure 1B, sucrose: F(1, 18) = 28.04, *p* < 0.0001, fat: F(1, 18) = 25.64, *p* < 0.0001; Appendix A, sucrose: F(1, 9) = 46.01, *p* < 0.0001; fat: F(1, 9) = 33.92, *p* = 0.0003). The results also indicate that, during the first hour of access, rats in the IA group consumed significantly more sucrose and fat compared to the second hour. In contrast, no significant differences were observed in sucrose or fat consumption between the first and second hours of access in the CA group (Appendix A, IA 1st vs. 2nd h, sucrose: *p* = 0.0006; fat: *p* = 0.0166). Total caloric intake was significantly higher in the CA and IA groups compared to the NA group, indicating sustained hyperphagia (Figure 1C, F(2, 31) = 14.64, *p* < 0.0001, NA vs. CA: *p* = 0.0155, NA vs. IA: *p* < 0.0001). Notably, rats in the CA group consumed more sucrose than those in the IA group (Figure 1D, CA vs. IA, *p* < 0.0001), while no significant difference was observed in total fat intake between these two groups (Figure 1C, CA vs. IA, *p* = 0.5910).

An intra-group comparison showed that rats in the IA group consumed significantly more calories from fat than from sucrose (Figure 1D, IA, sucrose vs. fat, *p* = 0.029); whereas, no difference was observed for sucrose and fat calorie intake among CA rats (Figure 1C, CA, sucrose vs. fat, *p* = 0.3053). In both cohorts, there were no significant difference between groups in weight progression throughout the diet or in final weight (Figure 1E, Appendix A). However, the CA group exhibited a greater amount of visceral white adipose tissue compared to control rats in both cohorts (Figure 1D, F(2, 31) = 7.99; CA vs. NA, *p* = 0.013; Appendix A, NA vs. CA, *p* = 0.0513), reflecting a hallmark feature of obesity.

### 2.2. Intermittent and Continuous Palatable Food Access Induced Differential Regulation of Endocannabinoid System Gene Expression

After the 6-week diet, the PCR analysis of ECS gene expression was conducted in reward-related brain regions, including the DS, NAc, VTA, and RMTg. In the DS, a significant downregulation of CB1 receptor expression was observed in the IA group (Figure 2A, F(2, 31) = 3.98, NA vs. IA, *p* = 0.0290), with no significant changes detected in other ECS receptors or enzymes in this region. In the NAc, the transcript levels of both CB1 and CB2 receptors were downregulated exclusively in the IA group (Figure 2B, CB1: F(2, 21) = 6.16, *p* = 0.0079, NA vs. IA, *p* = 0.0057; CB2: F(2, 23) = 6.21, *p* = 0.0070, NA vs. IA, *p* = 0.0050). Additionally, the expression of the 2-AG synthesis enzyme DAGL was significantly downregulated in both CA and IA groups compared to the NA group (Figure 2B, F(2, 21) = 11.39, *p* = 0.0004; NA vs. CA, *p* = 0.0287; NA vs. IA, *p* = 0.0005), with no significant changes in other enzymes. In the VTA, no significant changes in gene expression were detected; although, considerable variability was observed for certain target genes, such as CB2 and GPR55 (Figure 2C). In the RMTg, CB1 receptor transcript levels were significantly decreased in the CA group, while CB2 receptor transcript levels were significantly increased in the IA group, both relative to the NA group (Figure 2D, CB1: F(2, 19) = 3.68, *p* = 0.0447, NA vs. CA, *p* = 0.0356, CB2: F(2, 21) = 4.10, *p* = 0.0313, NA vs. IA, *p* = 0.0383). Additionally, DAGL expression was significantly increased in the IA group compared to the NA group (Figure 2E, F(2, 21) = 3.77, *p* = 0.0399, NA vs. IA, *p* = 0.0437).

### 2.3. AEA Levels Were Decreased Following Binge-like Behavior

Endocannabinoid levels were assessed using mass spectrometry, as previously described [30,34]. In the NAc, AEA levels were significantly decreased in the IA group compared to the NA group (Figure 2E left panel, F(2, 27) = 4.487, *p* = 0.0208, IA vs. NA, *p* = 0.0229). No significant differences in AEA levels were observed in the DS, and AEA was undetectable in both the VTA and RMTg. Regarding 2-AG levels, no significant changes were detected in any of the four targeted brain regions in our conditions. (Figure 2E right panel). Correlation analyses between gene expression and endocannabinoid levels were not performed, because these measures were obtained from two distinct cohorts (see Section 4).

### 2.4. Gene Co-Expression Network Comparison Analysis Revealed Differential Signatures Associated with Diet Access

The comparisons of correlation matrices (Figure 3A) using the Mantel test assessed the similarity in ECS gene expression correlation patterns across brain regions between the NA, CA, and IA groups. The comparison between NA and CA groups (Mantel statistic = 0.2531, *p* < 0.001) indicated a moderate and statistically significant similarity, suggesting shared features in their ECS gene expression correlation patterns, while also reflecting notable differences. Similarly, the NA vs. IA comparison (Mantel statistic = 0.2081, *p* < 0.001) demonstrated a moderate similarity, though the lower Mantel statistic compared to NA vs. CA suggested greater changes in correlation structures in the IA group, making it more distinct from the NA group. Interestingly, the CA vs. IA comparison (Mantel statistic = 0.1752, *p* < 0.001) revealed a weaker similarity, indicating fewer shared features and substantial differences likely arising from their distinct patterns of exposure to palatable food.

Significant gene–gene correlation heatmaps revealed distinct patterns of ECS gene co-expression across brain regions between the three experimental groups (Figure 3B). In the NA group (Figure 3B, left panel), significant correlations were sparsely distributed, indicating a balanced and stable coordination of ECS gene expression under control condition (no access to palatable food). For the CA group (Figure 3B, middle panel), a greater number of significant correlations was observed, particularly in the NAc and VTA, suggesting that continuous access to palatable food enhances the region-specific coordination of ECS gene expression. However, significant correlations observed in the NA group disappeared in the RMTg, suggesting the altered coordination of ECS genes in this brain region following continuous access to palatable food. In contrast, the IA group displayed a distinct pattern, characterized by a broader reorganization of significant positive and negative correlations (Figure 3B, right panel). Notably, increased positive correlations were observed in the RMTg, while increased negative correlations were prominent in the DS, reflecting a unique adaptive response to palatable food bingeing. These differences underscore the specific impact of palatable food access schedules on the co-regulation of ECS-related genes in reward-related brain regions.

Differential heatmaps (Figure 3C) revealed distinct changes in ECS gene co-regulation between the NA, CA, and IA groups. Each comparison revealed unique patterns in ECS gene co-expression strengths. The NA vs. CA comparison showed relatively sparse differences, suggesting that continuous access to palatable food induced moderate changes in ECS gene co-regulation compared to the NA group (Figure 3C, left panel). In contrast, the NA vs. IA comparison revealed more pronounced differences, indicating that palatable food bingeing leads to a greater reorganization of ECS gene co-regulation relative to the NA group. The increased prevalence of stronger correlations in the IA group suggested a more dynamic ECS adaptation in mesostriatal brain regions following bingeing (Figure 3C, middle panel). Interestingly, the CA vs. IA comparison revealed the most substantial differences among the group comparisons, highlighting the contrasting effects of continuous vs. intermittent access to palatable food in the targeted brain regions. Specifically, binge-like behavior in the IA group induced widespread changes in ECS gene co-regulation; whereas, continuous access in the CA group led to more selective adaptations (Figure 3C, right panel).

To further investigate ECS gene co-expression patterns across experimental conditions, co-regulation networks were constructed for each group using significant correlations between gene expressions (Figure 3D). The visual comparison of the networks revealed clusters with more nodes and were stronger in the CA (Figure 3D, middle panel) and IA (Figure 3D, right panel) groups compared to the NA group (Figure 3D, left panel), reflecting enhanced or altered ECS gene co-expression under different schedules of palatable food access. To identify the most characteristic genes for CA and IA groups, genes with the highest degree of connectivity and those with the highest betweenness centrality within each network were analyzed (see Methods) (Figure 3E). In the CA group, the most characteristic genes were NAPE in the striatal regions (DS and NAc) and CB1 in the VTA. For the IA group, the most characteristics genes were FAAH in the DS and RMTg (Figure 3E, left panel), as well as CB2 and DAGL in the NAc (Figure 3E, right panel).

To assess whether specific brain regions stood out under different experimental conditions (NA, CA, IA), the node degree metric was calculated for each gene–region combination within each network. Notably, in the IA group, the RMTg exhibited the highest sum of node degrees (Figure 3F), indicating increased connectivity within the ECS gene co-expression network compared to other brain regions in response to binge eating.

### 2.5. Correlation Analysis Revealed Distinct Impacts of Fat and Sugar Overconsumption on ECS Gene Expression

In the IA group, a negative correlation was observed in the DS between NAPE-PLD expression and total fat consumption (Figure 4A, R^2^ = 0.5114, *p* < 0.05). Conversely, in the CA group, MGL expression positively correlated with total sucrose intake (Figure 4B, R^2^ = 05137, *p* < 0.05) but negatively correlated with total fat consumption (Figure 4A, R^2^ = 0.6191, *p* < 0.05). In the NAc, a positive correlation was found between CB1 expression and total sucrose intake in the IA group (Figure 4B, R^2^ = 0.7364, *p* < 0.05). In the CA group, DAGL expression was positively correlated with total fat intake (Figure 4A, R^2^ = 0.9398, *p* < 0.01). No significant correlations were observed in the VTA. Additionally, NAPE-PLD and FAAH transcript expression positively correlated with total sucrose (Figure 4B, R^2^ = 0.7174, *p* < 0.01) and fat (Figure 4A, R^2^ = 0.6103, *p* = 0.0381) intake, respectively. Pearson’s correlation analysis in the DS revealed opposite relationships between AEA levels and total food intake in the CA and IA groups (Figure 4C, AEA vs. CA, R^2^ = 0.5660, *p* < 0.0121; AEA vs. IA, R^2^ = 0.4501, *p* < 0.0337).

## 3. Discussion

### 3.1. Main Findings

In the present study, our primary aim was to assess whether distinct eating patterns involving a combination of fat and sucrose differentially influence ECS expression by directly comparing rat models of obesity and BED. Using qPCR and mass spectrometry, we identified ECS expression profiles across groups with different palatable food access schedules, revealing region-specific variations in striatal and mesencephalic brain areas. These results highlight unique ECS signatures associated with each eating pattern, which are further discussed below.

### 3.2. Preclinical Models Highlighted Bingeing on Fat

Several animal models have been developed to study binge eating and food overconsumption [3,6] with the type of palatable food used in these models emerging as a critical factor [37,39]. Clinical data suggest that the combination of fat and sugar enhances the pleasure associated with palatable food [40], while reducing the appeal of low-fat alternatives [41]. The combination likely contributes to excessive food consumption and associated pathologies, such as obesity and BED. In this study, we used a combination of fat and sugar presented as distinct items, as previously described [7,30], to produce two distinct patterns of palatable food intake: intermittent and continuous access. Our findings demonstrate that these patterns resulted in distinct hallmark characteristic of each pathology. Under our experimental conditions, the CA group exhibited significantly higher overall caloric intake, which correlated with increased white adipose tissue mass compared to the NA and IA groups. This outcome aligns with a commonly used criterion for validating obesogenic diet in rodents [30,42] and corresponds to the WHO definition of obesity as abnormal and excessive fat accumulation. In the binge-like model, the IA group exhibited a significantly higher intake of both fat and sucrose during the 2 h access period compared to the CA group, with the majority of this consumption occurring during the first hour, a defining feature of bingeing behavior. These results suggest a binge eating phenotype in the IA group, characterized by high palatable food consumption within a short time frame, consistent with previous reports on intermittent palatable food access [33,43]. Notably, the IA group reached total fat calorie intake levels similar to the CA group, despite having access for only 2 h, 3 d/week, further supporting a bingeing behavior focused on fat. A potential limitation of this study is the absence of a control group with low-restricted consumption (2 h, 7d/week) to compare bingeing behavior more directly. Nevertheless, comparing the IA and CA groups enabled a robust evaluation of BED and obesity phenotypes, offering insights into the specific ECS regulations underlying these two pathologies, as discussed in the following sections.

### 3.3. Distinct Signatures of Palatable Food Access Schedules on Gene Expression Patterns in the Mesolimbic Brain

Statistical analyses (*p* < 0.001) revealed that gene expression correlation patterns across all comparisons were structured and not random, indicating shared relationships between groups. However, the moderate correlation values highlighted notable differences in the organization of these patterns under distinct conditions. The NA and CA groups exhibited moderate similarity, suggesting that continuous access to palatable food induced changes in gene correlation patterns, while retaining some resemblance to the control condition. In contrast, the lower similarity between NA and IA indicated that intermittent access led to a more pronounced reorganization of ECS gene correlation structures. The weakest correlation, observed between CA and IA, reflected the distinct impacts of each access to palatable food on the coordination of gene expression. These findings underscore that the schedule of palatable food access differentially affect the coordination of ECS gene expression across mesolimbic brain regions. The moderate similarity between NA and CA groups suggested that the schedule of palatable food access partially disrupts ECS gene co-regulations. In contrast, the IA group displayed a more dynamic reorganization of gene expression, likely reflecting unique behavioral and neurobiological adaptations to restricted palatable food availability, such as altered reward sensitivity or stress-related responses. Furthermore, the weak correlation between CA and IA highlighted how continuous versus intermittent access distinctly impacted the coordination of ECS gene coordination. These differences provide valuable insights into the mechanisms underlying compulsive overeating and binge eating, emphasizing the role of access schedules in shaping neurobiological responses to palatable food. Comparing gene co-regulation networks provided a comprehensive visualization of gene co-regulation patterns, highlighting significant clusters that may play crucial roles in the neurobiological response to different dietary conditions. The identification of gene clusters using a community detection approach provided insights into how continuous and intermittent access to palatable food uniquely affects ECS genes regulation across reward-related brain regions. Clusters with more nodes and stronger connections were observed in the CA and IA groups compared to the NA group, reflecting enhanced or altered ECS gene co-regulation under different schedules of palatable food access. Additionally, unique ECS gene co-regulation patterns and characteristic genes were identified for CA and IA groups, revealing signatures that may reflect specific adaptations of ECS gene expression networks in response to the respective feeding paradigms.

Regional connectivity analysis further highlighted the differences in brain region involvement in co-regulation networks. Notably, the RMTg emerged as a prominent player in the adaptive response to binge eating behavior. The high sum of node degrees observed in the IA group suggests that the RMTg becomes particularly active in coordinating ECS gene co-regulation when the animals have intermittent access to palatable food. The RMTg is known to play a role in dopamine systems inhibitory control and aversion (see Introduction). Its increased connectivity within the ECS gene network may reflect an enhanced role in processing reward prediction error or modulating aversive signals associated with intermittent access. Intermittent access to palatable food creates a heightened state of reward expectation, and the heightened activity of the RMTg within the ECS gene expression network may reflect its role in regulating the emotional and motivational consequences of this expectation, potentially bridging reward and aversion pathways.

### 3.4. Specific Regulations of the ECS in the Striatum

In the IA group with intermittent access to both fat and sucrose, we observed a specific decrease in CB1 expression in both ventral and dorsal striatum. These findings align with previous research, showing similar results in animals bingeing on fat. For instance, CB1 receptor mRNA expression was decreased in the NAc of juvenile mice bingeing on fat [44]. Similarly, autoradiography studies demonstrated an approximately 30% reduction in CB1 receptor density in the NAc shell of female rats exposed to repeated access to highly palatable food, under both intermittent and continuous access conditions [31]. In the same study, a roughly 20% reduction in CB1 mRNA was also observed in the cingulate cortex (areas 1 and 2), specifically in bingeing rats [31]. Another study found that margarine intake significantly reduced CB1 receptor density in the PFC (cingulate cortex Cg1 and Cg3), under both low-restricted (2 h/d every day) or high-restricted (2 h/d, 3d/week) access to fat compared to control rats [33]. Furthermore, studies have identified the vmPFC-NAc pathway as a critical neurocircuitry involved in the inhibitory control of both impulsivity and fat bingeing behavior [45]. Collectively, these results underscore the importance of cortico-striatal connectivity in binge eating behavior and highlight the critical role for CB1 receptors within this circuit.

On the other hand, studies focusing solely on sucrose consumption reported increased CB1 expression in the NAc following bingeing behavior. For instance, intermittent access to liquid sucrose (12 h/d over 4 weeks) in rats led to increased CB1 expression [34]. Similarly, a regimen of 2 h/day, 5 days/week access to sucrose over 5 weeks in mice also induced an increase in CB1 expression in the NAc [37]. These results suggest that distinct highly palatable diets differentially impact CB1 receptor expression in the NAc. In this line, we observed a positive correlation between total sucrose intake and CB1 receptor expression in the NAc within the IA group, indicating that sucrose is likely the key macronutrient driving CB1 overexpression. Similar results have been reported in other brain regions, like the hippocampus, where continuous access to sucrose or fructose led to increased CB1 expression [46], extending these findings to additional brain regions associated with reward processing.

Under our experimental conditions, CB2 receptor expression was also decreased in the NAc of the IA group. This finding supports a recent report highlighting the role of CB2 receptor in the development of “food addiction”. In that study, mice lacking CB2 receptor exhibited reduces compulsive behavior in an operant protocol with chocolate-flavored pellet delivery [47]. A lower percentage of these CB2 knockout mice developed food addiction during the early training period suggesting a resistant phenotype. Conversely, mice overexpressing the CB2 receptor displayed a higher vulnerability to food addiction during long-term exposure. Hence, the decreased CB2 receptor expression that we observed in the NAc could represent a neuroadaptive mechanism aimed at limiting palatable food intake under intermittent access condition. Further investigations, with prolonged exposure to palatable food, would help to disentangle key mechanisms underlying these molecular adaptations, particularly to determine whether CB2 receptor expression regulation persists over a longer regimen.

### 3.5. Specific Regulations in the RMTg

In the RMTg, we observed a reduction in CB1 receptor expression in the CA group, supporting prior evidence that CB1 signaling in this region modulates VTA dopaminergic activity [28,48,49]. This reduction may represent an adaptive mechanism aimed at mitigating the overactivation of the reward pathway and regulating palatable food intake. In contrast, binge eating (IA) rats exhibited increased ECS activity in the RMTg, including elevated DAGLα and CB2 expression, potentially reflecting the initiation of a progressive enhancement of 2-AG tone that could be responsible for DA neuron disinhibition and contribute to the vulnerability for compulsive palatable food intake. This mechanism aligns with studies linking GABAergic input modulation from the RMTg to increased palatable food consumption [50,51]. Interestingly, CB2-GPR55 correlations observed in the IA but not CA groups suggest distinct ECS adaptations depending on feeding patterns. While CA rats consumed fat consistently over six weeks, IA rats consumed similar quantities but in concentrated binge episodes, potentially hindering adaptive mechanisms and promoting further overconsumption.

Our results further highlight ECS expression differences between CA and IA groups, with reduced ECS activity in the RMTg following obesity protocol but increased activity following BED protocols. This is in accordance with evidence that elevated ECS tone in the RMTg enhances alcohol consumption [28] and suggests a similar role in palatable food intake. Importantly, the dissociative effects of fat versus sucrose on ECS expression underscore the need to distinguish their respective roles in dysregulation of reward pathways contributing to obesity and BED. Together, these findings emphasize the need for further research into RMTg-specific ECS modulation and its implications for obesity and BED. Further experiments will also be conducted to explore whether transcriptomic adaptations of the ECS observed in our study are detectable at the protein level.

## 4. Materials and Methods

### 4.1. Subjects

A total of 64 male Wistar rats (Janvier Labs, Le Genest-Saint Isle, France) weighing 250 ± 20 g at arrival were individually housed in standard cages under controlled 12 h light/dark cycle (lights on 07:00–19:00) and standard temperature and humidity conditions (22 ± 2 °C, 55 ± 10%). Two separate cohorts of rats were used for this study. All procedures involving animal care were conducted in compliance with current laws and policies validated by the CREMEAS (Comité d’Éthique pour l’Expérimentation Animale de Strasbourg) and approved by the Ministère de l’Enseignement Supérieur, de la Recherche et de l’Innovation (APAFIS #2019070816359145 v2). All efforts were made to minimize animal discomfort and to use the smallest number of rats necessary to achieve reliable results.

### 4.2. Free-Choice High-Fat High-Sucrose Diet (HFHS)

Rats from 2 distinct cohorts were exposed to the free-choice high-fat high-sucrose paradigm to explore its consequences on cannabinoid gene expression (cohort A) and endocannabinoid levels (cohort B) (Figure 1A). Following a 2-week habituation period to the animal facility, rats were divided into three groups. The control group (NA) was exposed to water and standard chow diet (SAFE A03, 3.339 kcal/g, 3. 16.1% of proteins, 3.1% of lipids, 3.9% of fibers, 4.6% of inorganic matter and 60.4% nitrogen-free extract, Rosenberg, Germany). The continuous free-choice high-fat high-sugar group (fcHFHS) (CA) was exposed to standard chow and water, combined with saturated fat (“blanc de bœuf” Paturage des Flandres, France, 9 kal/g) and sucrose (10% dissolved in water, Powder Sugar, Erstein, 4 kcal/g), as previously described [30]. The intermittent access (IA) group had access to the same palatable food for 2 h per day, three times a week (3 p.m. to 5 p.m., MWF: Monday, Wednesday, Friday). This regimen lasted for five weeks for cohort B (NA = 10; CA = 10; IA = 10) and six weeks for cohort A (NA = 14; CA = 10; IA = 10). Food and liquid intake were measured twice daily at 3 and 5 p.m., with additional measurement in cohort B during the first hour of palatable food access in weeks 4 and 5. Rats were weighed weekly, and total food intake was normalized to kilocalories per 100 g of body weight.

### 4.3. Gene Expression Analysis Using Quantitative Polymerase Chain Reaction (qPCR)

Animals were anesthetized with ketamine/xylazine (82.5 mg/kg/11 mg/kg, i.p.) and euthanized with an overdose of pentobarbital (140 mg/rat, i.p.). Epididymal white adipose tissue was dissected and weighed, and brain regions, including the DS, NAc, VTA, and RMTg, were bilaterally microdissected, as previously described [30]. A 1 mm slice corresponding to the RMTg was carefully isolated. Total RNA was extracted by homogenization of the sample in 0.8 mL of UpTizol RNA Extraction Reagent (Interchim©), and RNA quality and concentration were assessed with a NanoVue™ spectrophotometer (GE healthcare). Reverse transcription was performed using the iScript (iScript™ cDNA Synthesis Kit, Biorad, France) with 750 ng of total RNA for DS and NAc, 80 ng for the VTA, and 100 ng for the RMTg due to lower RNA yield in these structures. Real-time PCR was performed in triplicate using a CFX96 Touch™ apparatus (Biorad, France) and Sso Advanced™ Universal SYBR Green supermix (Biorad, France). The thermal cycling parameters were 30 s at 95 °C, followed by 40 cycles of amplification (5 s at 95 °C and 45 s at 60 °C). Primer sequences for all tested genes have been described previously, for Rplp0 (NM_022402), Cnr1 (NM_012784), Cnr2 (NM_001164143), Nape-Pld (NM_199381), Mgll (NM_138502), FAAH (NM_001369126) [30,34], DAGLα (XM8039079785.1) [52], and GPR55 (XM_006245493.4) [53]. Gene expression levels were normalized to Rplp0 housekeeping gene levels, commonly used in brain- or food-related studies [54,55]. Relative gene expression changes were calculated using the 2^−ΔΔCt^ method, comparing control and experimental samples [56].

### 4.4. Quantification of Endocannabinoid Levels by Mass Spectrometry

DS, NAc, VTA, and RMTg samples from cohort B were prepared, as previously described [30]. Briefly, tissues were sonicated (2 times 5 s, 90 W) in 200 µL of H_2_O. Then, homogenates were centrifuged (20,000× *g*, 30 min, 4 °C), and the resulting supernatants (150 µL) were mixed with 50 µL of 100% acetonitrile (ACN) containing 400.26 pmol of D8-2AG (sc-480539; Santa Cruz, Heidelberg, Germany) and 100.15 pmol of D4-AEA (Tocris/Biotechne, Lille, France). After another centrifugation step (20,000× *g* for 30 min, 4 °C), supernatants were collected and evaporated to dryness. Dried samples were re-suspended in 20 µL of ACN 30%/H_2_O 69.9%/formic acid 0.1% (*v*/*v*/*v*). Notably, AEA was only detectable in DS and NAc samples. Endocannabinoid levels were analyzed using a Dionex Ultimate 3000 HPLC system coupled with a triple quadrupole Endura mass spectrometer (Thermo Scientific, San Jose, CA, USA), controlled by Xcalibur v. 2.0 software (Thermo Electron, San Jose, CA, USA). Samples (3 µL/20 µL) were loaded onto the column (BASi Unijet microbore 1 mm × 10 mm, 3 µm; Bioanalatical Systems, Lafaiyette, LA, USA) heated at 40 °C. Elution was performed by applying a gradient of mobile phases A/B (flow rate of 50 µL/min). Qualification and quantification were performed in the multiple reaction monitoring mode (MRM) and based on precursor ion, selective fragment ions, and retention times obtained for 2-AG, AEA, D8-2-AG, and D5-AEA (details provided in [30]). All endocannabinoid levels fell within standard curve limits, with analytical ranges typically spanning from 1 fmol/100 pmol to 150 fmol/100 pmol. Precision (CV% between repeated injections of the same sample) values were <1% for same-day measurements and <5% for interday measurements.

### 4.5. Statistical Analysis

All results are expressed as mean ± SEM. Statistical analyses were performed using GraphPad Prism 10.4 (GraphPad Software, La Jolla, CA, USA), R-studio 4.0.2, and Python 3.8. All datasets were tested for normality and homoscedasticity before applying parametric statistical tests. Total food intake was analyzed using a one-way-ANOVA. Body weight progression and weekly food consumption were analyzed with a two-way ANOVA with repeated measures or mixed-effects analysis in case of missing data. Tukey’s multiple comparisons were performed to assess group differences over time. Gene expression analyses were performed with one-way ANOVA followed by Tukey’s post hoc. Outliers for qPCR analysis were identified when ∆Cq and relative gene expression values were outside the [Q1 −1.5 I QR; Q3 +1.5 IQR] range. Correlation and linear regression analysis were performed either to compare sucrose/fat intake over time to gene expression or in between genes relative expression. The comparison of sucrose or fat intake during the 1st vs. 2nd h was analyzed with paired (intra-group comparisons) or unpaired (inter-group comparisons) Student’s *t*-test.

### 4.6. Data Preprocessing and Correlation Analysis

Data imputation. Missing values in the gene expression dataset were imputed using the K-nearest neighbors (KNN) algorithm with n_neighbors = 5. Correlation matrices. For each group (NA, CA, IA), correlation matrices were computed using the Pearson correlation coefficient to represent pairwise correlations between gene expression levels across brain regions. To evaluate the similarity of gene expression correlation structures between groups, the Mantel test was applied, with statistical significance assessed via permutation-based empirical *p*-values. A Bonferroni correction was applied to control for multiple comparisons, with a significance threshold of 0.05. Only meeting this criterion were retained. Differential correlation matrices. For each pair of groups (NA vs. CA, NA vs. IA, CA vs. IA), correlation matrices were subtracted from one another to obtain a differential matrix. For each element of the matrix (i.e., for each pair of genes), the correlation value in group 1 was subtracted from the corresponding value in group 2. Network creation. Each gene–region combination was represented as a node, with edges added between nodes if a significant correlation (after Bonferroni correction) was observed. The networks were drawn using the spring layout algorithm, which positions nodes in a manner that minimizes edge crossings and results in a visually intuitive layout. Community detection and clustering. To identify clusters of co-regulated genes in each network, we used the greedy modularity communities’ algorithm (NetworkX 3.1 package), which aims to optimize modularity to detect communities or clusters within the network. This algorithm iteratively finds sets of nodes that are more densely connected internally compared to connections with nodes outside the set, effectively identifying functional modules of co-regulated genes. Each cluster identified by the greedy modularity communities’ algorithm (NetworkX 3.1 package) was assigned a distinct color. Visualization of characteristic genes. Key genes within the CA and IA groups were identified based on degree of connectivity and betweenness centrality within each network. Gene–region pairs in the top 10% for both metrics were classified as key nodes and highlighted in red. Regional connectivity analysis. Node degree and betweenness centrality analysis were performed at the brain region level (DS, NAc, VTA, and RMTg). Metrics for each brain region were summed across all genes within that region, providing an aggregate measure of connectivity and regulatory importance for each brain area.

## 5. Conclusions

Together, our findings underscore the dynamic nature of the mesostriatal ECS response to continuous versus intermittent access to palatable food, revealing distinct patterns of co-regulation across brain regions that may drive behavioral adaptations related to reward-seeking, motivation, and emotional processes. These insights open avenues for exploring the molecular mechanisms underlying adaptive behaviors in fluctuating environments of palatable food availability and identifying novel ECS targets for the treatment of obesity and binge eating disorder.

## Figures and Tables

**Figure 1 ijms-26-01240-f001:**
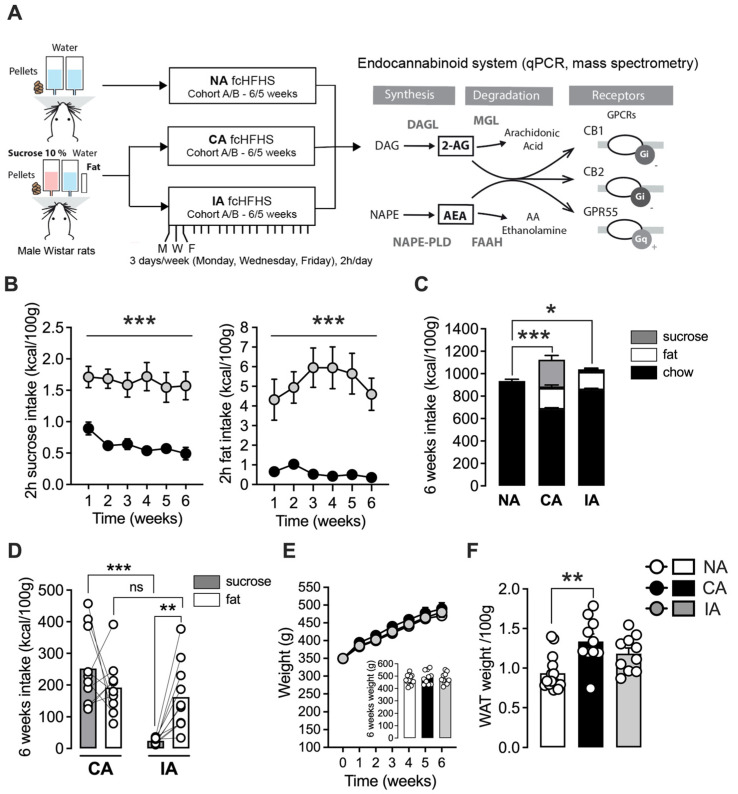
Free-choice high-fat high-sucrose as a model of obesity and binge eating. (**A**) Timeline of the diet exposure and experimental procedures. Following diet exposure (see Methods for details), the expression of the different ECS components was assessed in samples from cohort A (6 weeks), using qPCR and in samples from cohort B (5 weeks) using mass spectrometry. (**B**–**F**) Data from cohort A. (**B**) The intermittent access (IA) to palatable food induced a higher fat and sucrose intake during the 2 h access throughout the entire duration of the protocol, in comparison with the continuous access group (cohort A). (**C**) Rats in the CA and IA groups have an overall higher caloric intake than the NA group. (**D**) Total amount of sucrose intake is higher in the CA group in comparison with the IA group; whereas, the total amount of fat is similar in both groups. Rats in the IA group consumed more fat than sucrose. (**E**) Weight gain throughout the diet was similar between groups. (**F**) The CA group showed an increased adiposity. Data for cohort B are presented in Appendix A. NA: no access (n = 10–14), CA: continuous access (n = 8–10), IA: intermittent access (n = 10). n= number of rats. Results are presented in mean ± SEM. * *p* < 0.05; ** *p* < 0.01; *** *p* < 0.001.

**Figure 2 ijms-26-01240-f002:**
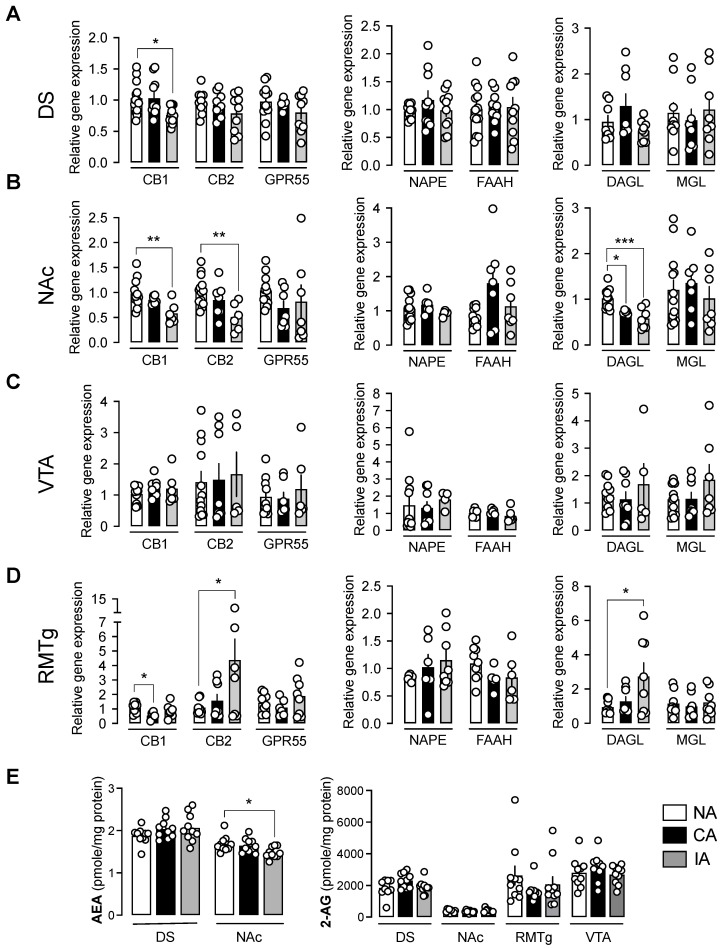
Intermittent and continuous access to fat and sucrose modulated endocannabinoid system expression in the striatum and the RMTg, with greater effect in the IA group. (**A**) In the dorsal striatum (DS), CB1 gene expression was significantly decreased only in the IA group. (**B**) CB1, CB2, and DAGLα gene expression were significantly decreased in the NAc of rats in the IA group, and DAGLα gene expression was also decreased in the CA group. (**C**) Gene expression for the endocannabinoid system in the VTA was not significantly regulated following intermittent or continuous access to palatable food. (**D**) In the RMTg, the expression of CB1 was significantly reduced in the CA group; whereas, CB2 and DAGLα gene expression were increased in the IA group. (**E**) AEA levels were downregulated in the IA group in the NAc, and they were not detectable in VTA and RMTg (left panel). No significant regulation of 2-AG levels was observed in the four targeted brain structures (right panel). NA: no access (n = 10–14), CA: continuous access (n = 6–10), IA: intermittent access (n = 4–10). Results are expressed as mean ± SEM. * *p* < 0.05; ** *p* < 0.01; *** *p* < 0.001.

**Figure 3 ijms-26-01240-f003:**
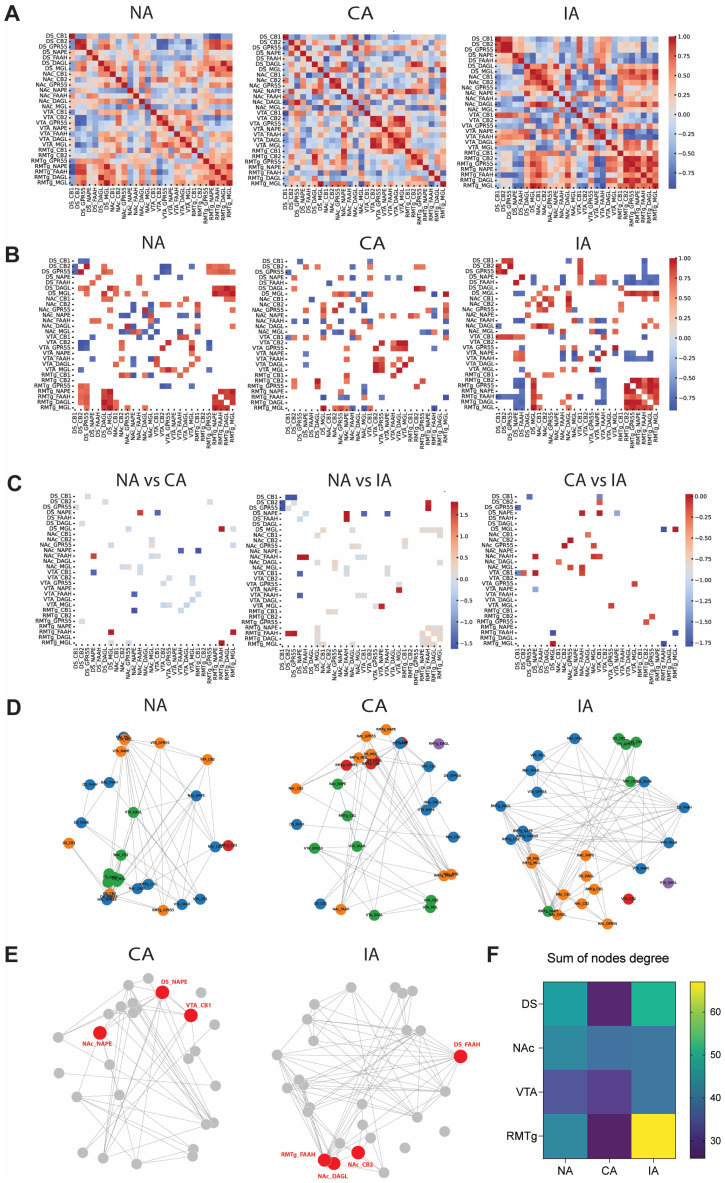
Distinct ECS gene co-regulation patterns and enhanced RMTg connectivity in the IA group highlighted the impact of palatable food access schedules. (**A**) Heatmaps showing Pearson correlation coefficients between ECS genes across brain regions for each group: NA, left panel, CA, middle panel, and IA right panel. Positive correlations are shown in red, and negative correlations in blue, with the intensity of the color indicating the strength of the correlation. (**B**) Heatmaps displaying only significant Pearson correlation coefficients (Bonferroni-corrected) between ECS genes across investigated brain regions. Differences in the pattern and distribution of significant correlations highlighted the distinct effects of continuous and intermittent access to palatable food on ECS gene co-regulation. (**C**) Differential heat maps displaying differences in significant ECS gene co-expression between groups: NA vs. CA (left panel), NA vs. IA (middle panel), and CA vs. IA (right panel). Each cell represents the difference in correlation values for a specific pair of genes across brain regions. Positive values (red) indicate stronger correlations in the first group compared to the second group, while negative values (blue) indicate stronger correlations in the second group compared to the first one. Grey cells indicate no significant difference. Differences highlighted the unique effects of binge eating on ECS gene co-regulation in investigated brain regions. (**D**) Co-regulation networks of ECS gene expression across the three groups: NA, left panel, CA, middle panel; IA, right panel. Each node represents a gene–region pair, and edges indicate significant correlations between them. Clusters of co-regulated genes, represented by different colors, highlighted distinct modules of coordinated activity within each group. (**E**) Visualization of characteristic genes for CA and IA group revealing central ECS gene components within each network. Nodes with a high number of connections and a high betweenness centrality (top 10%) were identified as key genes in the network and highlighted in red. (**F**) Visualization of summed node degrees for regional connectivity comparison across conditions reveals a higher RMTg connectivity in the ECS gene co-expression network in the IA group.

**Figure 4 ijms-26-01240-f004:**
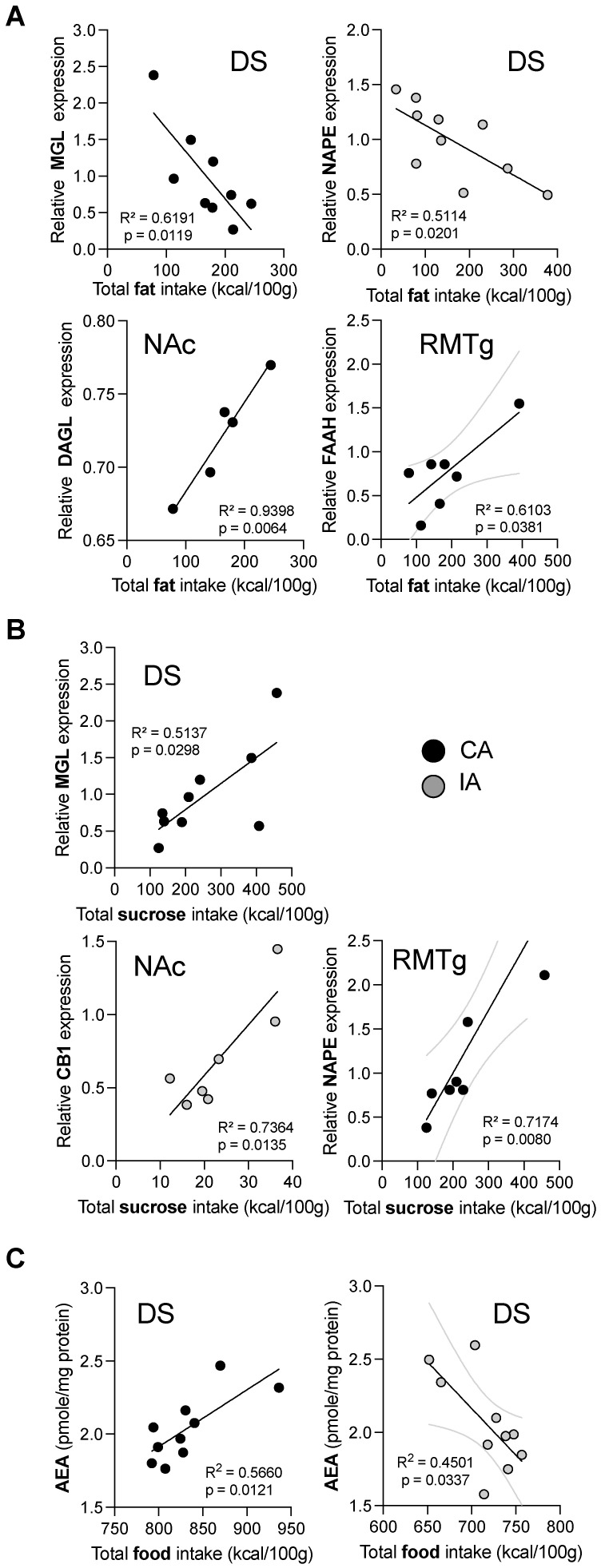
Correlation analysis showed a different impact of fat vs. sugar consumption on the endocannabinoid system gene expression. (**A**) Correlation with fat intake: In the DS, a negative correlation was observed between NAPE expression and total fat intake in the IA group. In the CA group, MGL expression negatively correlated with total fat intake in the DS. In contrast, positive correlations between DAGL or FAAH and total fat intake were observed in the CA group in the NAc and the RMTg. (**B**) Correlations with sucrose intake: In the CA group, positive correlations were observed between MGL and total sucrose in DS and NAPE and total sucrose in the RMTg. In the NAc, CB1 gene expression was positively correlated with total sucrose intake for the IA group. (**C**) Correlations with total food intake: In the CA group a positive correlation between AEA and total food intake was observed in the DS; whereas, this correlation was negative for the IA group. NA: no access (n = 10–14), CA: continuous access (n = 6–10), IA: intermittent access (n = 4–10). Results were presented as individual data points with the best fitted line and the 95% confidence interval bands.

## Data Availability

All the data supporting the findings of this study can be provided upon request from the corresponding author.

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
