# Peer review of "Binge Eating and Obesity Differentially Alter the Mesolimbic Endocannabinoid System in Rats"

_ijms, 2025, doi:10.3390/ijms26031240_

Round 1

Reviewer 1 Report

Comments and Suggestions for Authors

The study "Binge eating and obesity differentially alter the mesolimbic endocannabinoid system in rats" examined the effects of binge eating disorder (BED) and obesity on endocannabinoid system (ECS) expression in male Wistar rats. The researchers employed a six-week experimental design with three groups: a non-access (NA) group fed a standard diet, a continuous access (CA) group with unrestricted access to a high-fat, high-sugar diet to model obesity, and an intermittent access (IA) group with periodic access to the same diet to simulate BED. Food consumption was monitored throughout the study. Upon completion, the researchers collected brain samples to analyze ECS expression in key regions, including the nucleus accumbens (NAc), dorsal striatum (DS), ventral tegmental area (VTA), and rostromedial tegmental nucleus (RMTg). The analysis was conducted using quantitative PCR (qPCR) and mass spectrometry techniques.   This study's findings shed light on the complex interplay between diet composition and the endocannabinoid system (ECS) across various brain regions. Furthermore, the observed co-regulation patterns of ECS-related genes suggest that distinct neural circuits may be involved in mediating the behavioral and physiological responses to different types of palatable foods. The article objective and experimental methodology fit the study justification perfectly. The results are good and suitable, and the discussion is well written.

Minor comment:

To strengthen the study's robustness, it is recommended that the authors include protein expression data of ECS expression using western blotting.

Author Response

Q1.Minor comment:

To strengthen the study's robustness, it is recommended that the authors include protein expression data of ECS expression using western blotting.

We do understand the comment of the reviewer, as protein regulation would bring a great addition to our study. The aim of our study was to provide a transcriptional analysis in our two feeding paradigms mimicking obesity and binge eating disorder.

  • Unfortunately, all the tissues that were collected in these experiments have been used for either the qPCR or the mass spectrometry experiments to investigate transcriptomic and endocannabinoid regulations, respectively. We do not have any tissues left from the two cohorts. Therefore, addressing the issue thoroughly would need further study on a whole new cohort, which would involve a minimum of 30 additional rats and at least 6 months of work.
  • Moreover, for this purpose, we would need first to apply for a new authorization to the French minister for this specific animal experiment, as the authorization to work with animals that was obtained for the initial work is no longer running. This will further delay the starting of the experiment by about 3 months.
  • In addition, we would have to optimize the conditions for western blotting for several cannabinoid proteins to target, to make sure the antibodies are selective, as we do not run these routinely in our laboratory, which also represents a huge amount of work.
  • Adding to this, the first author Florian Schoukroun is now conducting a post-doctoral training in the USA since spring 2024 and would therefore not be available to complete these experiments. In addition, the funding for this project was concerning the transcriptomic analysis of the ECS following the regimens and protein analysis was not in the scope of the study. We therefore do not have additional funding so far to be able to complete these data at the protein level in the coming months.

We hope that both the reviewer and the editor would understand all these limitations for us to provide additional data in a reasonable time frame (>9 months). As we agree with the reviewer’s recommendation on this matter, we have added a statement in the discussion to underline the need for further research concerning protein regulation (line 441).

Further experiments will also be conducted to explore whether transcriptomic adaptations of the ECS observed in our study are detectable at the protein level.

Reviewer 2 Report

Comments and Suggestions for Authors

This research paper investigates the changes in the expression of the endocannabinoid system (ECS) in response to different eating patterns. Using a free-choice high-fat high-sugar (fcHFHS) diet model, the authors demonstrate changes in ECS gene expression in reward-related brain regions, such as the NAc, VTA, RMTg, and DS, and explore the relationship between these changes and eating patterns. Furthermore, through network analysis, the authors reveal that continuous and intermittent food access have significantly different effects on the co-regulatory network of ECS genes, with a particular focus on the relationship between the RMTg region and binge eating behavior. These findings provide a theoretical foundation for the future development of ECS-based intervention strategies, with notable academic and clinical implications.

However, there are several areas in which the authors should consider further revisions. The specific revisions are listed below:

1. In the introduction, the authors describe how the endocannabinoid system plays a role in binge eating and obesity, but the specific mechanisms are unclear. It is suggested that the author further elucidate the mechanism of endocannabinoid system under physiological and pathological conditions of other organs. The following references are recommended:

(1) Li L, Gao P, Tang X, et al. CB1R-stabilized NLRP3 inflammasome drives antipsychotics cardiotoxicity. Signal Transduct Target Ther. 2022; 7(1): 190.Published 2022 Jun 24. doi:10.1038/s41392-022-01018-7

(2) Zhou S, Wu Q, Lin X, et al. Cannabinoid receptor type 2 promotes kidney fibrosis through orchestrating β-catenin signaling. Kidney Int. 2021; 99 (2) : 364-381. The doi: 10.1016 / j.k. int. J 2020.09.025

2. The author's description of Figure legend 1 is not clear enough. The description in Figure 1D appears to be 1E and 1F. Please re-describe the figure legend for C, D, E, F in Fig. 1

3. Please revise the symbols in Figure 2E that do not show significant differences.

4. Please revise the incorrect "Table1" in the title of Figure 2.

5. In line 236, the description of "Fig.3F" appears, but there is no figure F in Fig.3. The author is needed to revies it.

6. In lines 239-270, the figure legend of figure 3 is too length. The section on the interpretation of the results could be placed in the conclusion and discussion to make the article more concise.

7. In line 278, it is mentioned that "In the CA group, a strong correlation was observed between the CB2 and GPR55 receptor transcript levels", However, there seems to be no corresponding result in Fig.4B to confirm this. Please confirm the correlation between.

Author Response

Q1. In the introduction, the authors describe how the endocannabinoid system plays a role in binge eating and obesity, but the specific mechanisms are unclear. It is suggested that the author further elucidate the mechanism of endocannabinoid system under physiological and pathological conditions of other organs. The following references are recommended:

(1) Li L, Gao P, Tang X, et al. CB1R-stabilized NLRP3 inflammasome drives antipsychotics cardiotoxicity. Signal Transduct Target Ther. 2022; 7(1): 190.Published 2022 Jun 24. doi:10.1038/s41392-022-01018-7

(2) Zhou S, Wu Q, Lin X, et al. Cannabinoid receptor type 2 promotes kidney fibrosis through orchestrating β-catenin signaling. Kidney Int. 2021; 99 (2) : 364-381. The doi: 10.1016 / j.k. int. J 2020.09.025

We initially selected only the most pertinent references to support our points on the role of the ECS in food intake in the central nervous system. We thank the reviewer for providing these references to point that the peripheral system expressing the ECS may also play a role in BED and obesity. We have added a sentence to specifically address this point and we preferred to use a review (ref#38) describing the ECS modulation in peripheral organs in physiological and pathological conditions to be more exhaustive. See line 90, text in blue:

It is important to note that modulation of the ECS may also be impacted by peripheral signals, as this system is altered in various peripheral pathologies, including metabolic diseases such as obesity, insulin resistance, type 2 diabetes, and cardiovascular diseases (5).

Q2. The author's description of Figure legend 1 is not clear enough. The description in Figure 1D appears to be 1E and 1F. Please re-describe the figure legend for C, D, E, F in Fig. 1

We thank the reviewer for pointing this problem. The legend has been corrected to fit each panel. Changes are indicated in blue in the § line 125.

Figure 1. Free-choice high-fat high-sucrose as a model of obesity and binge eating. (A) Timeline of the diet exposure and experimental procedures. Following diet exposure (see Methods for details), expression of the different ECS components was assessed in samples from cohort A (6 weeks) using qPCR and in samples from cohort B (5 weeks) using Mass spectrometry. (B-F) Data from cohort A. (B) The intermittent access (IA) to palatable food induced a higher fat and sucrose intake during the 2h access throughout the entire duration of the protocol, in comparison with the continuous access group. (C) Rats in the CA and IA groups have an overall higher caloric intake than the NA group. (D) Total amount of sucrose intake is higher in the CA group in comparison with the IA group whereas the total amount of fat is similar in both groups. Rats in the IA group consumed more fat than sucrose. (E) Weight gain throughout the diet was similar between groups. (F) The CA group showed an increased adiposity compared to the NA group. Data for cohort B are presented in SupFig1. NA : No-access (n = 10-14), CA : Continuous access (n = 8-10), IA : Intermittent Access (n=10). Results are presented in mean ± SEM.  * p < 0.05 ; ** p < 0.01 ; *** p < 0.001.

Q3. Please revise the symbols in Figure 2E that do not show significant differences.

 We thank the reviewer for pointing this problem. The symbol has been corrected on the panel 2E.

Q4. Please revise the incorrect "Table1" in the title of Figure 2.

We do apologize, but again, it seems that a formatting error has occurred when the template file was further sent to the reviewers. Our submitted version does not show this error.

The title of figure 2 was:  Figure 2. Intermittent and continuous access to fat and sucrose modulated endocannabinoid system expression in the stiatum and the RMTg, with greater effect in the IA group.

This has been now corrected in the revised version, line 171.

In addition, it seems that this error also deleted the beginning of the caption. Here is the complete text for legend of Figure 2 (line 171):

(A) In the dorsal striatum (DS), CB1 gene expression was significantly decreased only in the IA group. (B) CB1, CB2 and DAGLα gene expression were significantly decreased in the NAc of rats in the IA group, and DAGLα gene expression was also decreased in the CA group. (C) Gene expression for the endocannabinoid system in the VTA was not significantly regulated following intermittent or continuous access to palatable food. (D) In the RMTg, the expression of CB1 was significantly reduced in the CA group whereas CB2 and DAGLα gene expression were increased in the IA group. (E) AEA levels were down regulated in the IA group in the NAc and they were not detectable in VTA and RMTg (left panel). No significant regulation of 2-AG levels was observed in the four targeted brain structures (right panel). NA : No-access (n = 10-14), CA : Continuous access (n = 6-10), IA : Intermittent Access (n = 4-10). Results are expressed as mean ± SEM.  * p < 0.05 ; ** p < 0.01 ; *** p < 0.001.

Q5. In line 236, the description of "Fig.3F" appears, but there is no figure F in Fig.3. The author is needed to revies it.

We do apologize, the F letter for the corresponding panel has been added in Figure 3 (line 243).

Q6. In lines 239-270, the figure legend of figure 3 is too length. The section on the interpretation of the results could be placed in the conclusion and discussion to make the article more concise.

We have reduced the legend by deleting the interpretation that was already well developed in the result/discussion sections. Changes are indicated in blue, in the § line 243.

 Figure 3. Distinct ECS gene co-regulation patterns and enhanced RMTg connectivity in the IA group highlighted the impact of palatable food access schedules. (A) Heatmaps showing Pearson correlation coefficients between ECS genes across brain regions for each group: NA, left panel, CA, middle panel, and IA right panel. Positive correlations are shown in red, and negative correlations in blue, with the intensity of the color indicating the strength of the correlation. The visual comparison of heatmaps revealed distinct patterns of ECS gene co-regulation across groups. The NA group showed a balanced and sparse distribution of correlations (left panel), whereas the CA group exhibited more concentrated patterns (middle panel). In contrast, the IA group displayed a broader reorganization, reflecting the distinct impact of bingeing on gene expression coordination. (B) Heatmaps displaying only significant Pearson correlation coefficients (Bonferroni-corrected) between ECS genes across investigated brain regions. Differences in the pattern and distribution of significant correlations highlighted the distinct effects of continuous and intermittent access to palatable food on ECS gene co-regulation. The NA group showed sparse and balanced correlations, the CA group exhibited stronger region-specific coordination, and the IA group displayed broader reorganization, highlighting the impacts of each palatable food access schedule. (C) Differential heat maps displaying differences in significant ECS gene co-expression between groups: NA vs. CA (left panel), NA vs. IA (middle panel), and CA vs. IA (right panel). Each cell represents the difference in correlation values for a specific pair of genes across brain regions. Positive values (red) indicate stronger correlations in the first group compared to the second group, while negative values (blue) indicate stronger correlations in the second group compared to the first one. Grey cells indicate no significant difference. Differences highlighted the unique effects of binge eating on ECS gene co-regulation in investigated brain regions. (D) Co-regulation networks of ECS genes expression across the three groups: NA, left panel, CA, middle panel; IA, right panel. Each node represents a gene-region pair, and edges indicate significant correlations between them. Clusters of co-regulated genes, represented by different colors, highlighted distinct modules of coordinated activity within each group. Notably, the network for the IA group appeared more interconnected, with denser and more extensive co-regulation compared to the NA and CA groups. These differences suggested distinct patterns of gene co-regulation influenced by palatable food schedule of access. (E) Visualization of characteristic genes for CA and IA group revealing central ECS gene components within each network. Nodes with a high number of connections and a high betweenness centrality (top 10%) were identified as key genes in the network and highlighted in red. (F) Visualization of summed node degrees for regional connectivity comparison across conditions reveals a higher RMTg connectivity in ECS gene co-expression network in the IA group.

Q7. In line 278, it is mentioned that "In the CA group, a strong correlation was observed between the CB2 and GPR55 receptor transcript levels", However, there seems to be no corresponding result in Fig.4B to confirm this. Please confirm the correlation between.

The correlation information between CB2 and GPR55 is visible in the heatmap of Figure 3 (panel B). In Figure 4, we have chosen to only focus on gene vs palatable food correlations. To clarify, we have therefore deleted the information of the receptor/receptor correlation both in the text and in the legend of Figure 4.

Text, line 270:

In the IA group, a negative correlation was observed in the DS between NAPE-PLD expression, and total fat consumption (Fig. 4A, R² = 0.5114, p < 0.05). Conversely, in the CA group, MGL expression in the DS positively correlated with total sucrose intake (Fig. 4B, R²=05137, p<0.05) but negatively correlated with total fat consumption (Fig. 4A, R²=0.6191, p<0.05). In the NAc, a positive correlation was found between CB1 expression and total sucrose intake in the IA group (Fig. 4B, R²=0.7364, p<0.05). In the CA group, a strong correlation was observed between the CB2 and GPR55 receptor transcript levels (Fig. 4B, R²=0.6925, p<0.05), while DAGL expression was positively correlated with total fat intake (Fig. 4A, R²=0.9398, p<0.01). No significant correlations were observed in the VTA. However, in the RMTg, a positive correlation between CB2 and GPR55 levels was observed for the IA group (Fig. 4C, R²=0.8560, p<0.01). Additionally, NAPE-PLD and FAAH transcript expression positively correlated with total sucrose (Fig. 4B, R²=0.7174, p<0.01) and fat (Fig. 4A, R²=0.6103, p=0.0381) intake, respectively.

New Legend Figure 4, line 281.

Figure 4. Correlation analysis showed a different impact of fat vs. sugar consumption on the endocannabinoid system gene expression. (A) Correlation with fat intake: In the DS, a negative correlation was observed between NAPE expression and total fat intake in the IA group. In the CA group, MGL expression negatively correlated with total fat intake in the DS. In contrast, positive correlations between DAGL or FAAH and total fat intake were observed in the CA group in the NAc and the RMTg. (B) Correlations with sucrose intake: In the CA group, positive correlations were observed between MGL and total sucrose in DS, and NAPE and total sucrose in the RMTg. In the NAc, CB1 gene expression was positively correlated with total sucrose intake for the IA group. (C) Correlations with total food intake: In the CA group a positive correlation between AEA and total food intake was observed in the DS, whereas this correlation was negative for the IA group. NA : No-access (n = 10-14), CA : Continuous access (n = 6-10), IA : Intermittent Access (n = 4-10). Results were presented as individual data points with the best fitted line and the 95% confidence interval bands.

References

  1. Bourdy R, Hertz A, Filliol D, Andry V, Goumon Y, Mendoza J et al. The endocannabinoid system is modulated in reward and homeostatic brain regions following diet-induced obesity in rats: a cluster analysis approach. Eur J Nutr 2021.
  2. de Sa Nogueira D, Bourdy R, Filliol D, Awad G, Andry V, Goumon Y et al. Binge sucrose-induced neuroadaptations: A focus on the endocannabinoid system. Appetite 2021; 164: 105258.
  3. MacDowell KS, Sayd A, Garcia-Bueno B, Caso JR, Madrigal JLM, Leza JC. Effects of the antipsychotic paliperidone on stress-induced changes in the endocannabinoid system in rat prefrontal cortex. The world journal of biological psychiatry : the official journal of the World Federation of Societies of Biological Psychiatry 2017; 18(6): 457-470.
  4. Hurst K, Badgley C, Ellsworth T, Bell S, Friend L, Prince B et al. A putative lysophosphatidylinositol receptor GPR55 modulates hippocampal synaptic plasticity. Hippocampus 2017; 27(9): 985-998.
  5. Shrestha N, Cuffe JSM, Hutchinson DS, Headrick JP, Perkins AV, McAinch AJ et al. Peripheral modulation of the endocannabinoid system in metabolic disease. Drug discovery today 2018; 23(3): 592-604.
